# A 16S RNA Analysis of Yangzhou Geese with Varying Body Weights: Gut Microbial Difference and Its Correlation with Body Weight Parameters

**DOI:** 10.3390/ani14142042

**Published:** 2024-07-12

**Authors:** Xinlei Xu, Suyu Fan, Hao Wu, Haoyu Li, Xiaoyu Shan, Mingfeng Wang, Yang Zhang, Qi Xu, Guohong Chen

**Affiliations:** 1Key Laboratory for Evaluation and Utilization of Poultry Genetic Resources of Ministry of Agriculture and Rural Affairs, Yangzhou University, Yangzhou 225009, China; xuxinlei0914@163.com (X.X.); fansuyu20030729@163.com (S.F.); wh18755049672@163.com (H.W.); m19962648891@163.com (H.L.); lsshanxiaoyu218@163.com (X.S.); 18361310771@163.com (M.W.); xuqi@yzu.edu.cn (Q.X.); ghchen2019@yzu.edu.cn (G.C.); 2Joint International Research Laboratory of Agriculture and Agri-Product Safety, The Ministry of Education of China, Yangzhou University, Yangzhou 225009, China

**Keywords:** cecal microbiota, diversity, 16S rRNA, Yangzhou goose

## Abstract

**Simple Summary:**

China is a major goose-raising country, and the geese industry plays a significant role in animal husbandry. Goose growth performance (body weight) is a critical topic. This study used 16S rRNA sequencing to comprehensively analyze the cecal flora of healthy 70-day-old Yangzhou geese with varying weights but identical feeding methods to investigate the correlation between their intestinal microorganisms and body weight. Correlation analysis of production performance indicators between cecal microflora and body weight revealed a significant positive correlation between birth weight and *Deferribacterota* at the *phylum* level. At the genus level, leg and chest muscle weights exhibited significant positive correlations with *Prevotellace-ae_Ga6A1_group*, suggesting its critical role in promoting the growth and development of goose leg and chest muscles. These findings offer a crucial theoretical foundation for studying gastrointestinal microorganisms and provide insights into the development and formulation of poultry probiotics.

**Abstract:**

China is a major goose-raising country, and the geese industry plays a significant role in animal husbandry. Therefore, goose growth performance (body weight) is a critical topic. Goose gut microbiota influences weight gain by regulating its energy metabolism and digestion. Additionally, the impact of cecal microbial community structure on goose growth and development, energy metabolism, and immunity has been examined. However, most studies have used different additives or feeds as variables. Improving the understanding of the dynamic changes in gut microbial communities in geese of different body weights during their growth and development and their correlation with the host’s body weight is necessary. In this study, the cecal microbiota of healthy Yangzhou geese with large (L) and small (S) body weights, all at the same age (70 days old) and under the same feeding conditions, were sequenced using 16S rRNA. The sequencing results were annotated using QIIME2 (classify-sklearn algorithm) software, and the linkET package was used to explore the correlation between intestinal microorganisms and the body weight of the Yangzhou goose (Spearman). At the *phylum* level, the *Firmicutes*/*Bacteroidetes* ratio in the large body weight group was approximately 20% higher than that in the small body weight group, with *Bacteroidetes* and *Firmicutes* exhibiting a highly significant negative correlation. At the *genus* level, *Bacteroides* constituted the most abundant microbial group in both groups, although the *Prevotellaceae_Ga6A1_group* exhibited a higher abundance in the large than the small weight group. Spearman correlation analysis and the linkET package were used to analyze the correlation between cecal microflora and production performance indicators that showed significant differences between the two groups and showed that birth weight was significantly positively correlated with *Deferribacterota* at the phylum level. At the genus level, leg and chest muscle weights exhibited significant positive correlations with *Prevotellace-ae_Ga6A1_group*, suggesting its critical role in promoting the growth and development of goose leg and chest muscles. A significant negative correlation was observed between *[Ruminococ-cus]_torque* and *Prevotellaceae_Ga6A1_group*. These findings offer a crucial theoretical foundation for the study of gastrointestinal microorganisms and provide insights into the development and formulation of poultry probiotics.

## 1. Introduction

The intestinal microbiota of poultry is a vital component of the host physiological system, significantly affecting metabolism, immunity, health maintenance, and growth and development. Studies have shown that Lactobacillus, *Blautia bifobacterium*, *Faecalibacterium*, *Clostridium XlVa*, and members of the *phylum Firmicutes* are crucial for preventing and controlling avian influenza and other infections [1]. *Bacillus subtilis* and *Escherichia coli exert* beneficial effects on host animals by creating an intestinal microenvironment conducive to the inhibition of potentially pathogenic microorganisms [2,3]. *Macellibacteroides*, *Bacillus subtilis*, and *spore-forming bacteria* may be involved in life processes related to changes in duck carcass traits, such as variations in liver weight, abdominal fat weight, and abdominal fat percentage. Additionally, research has indicated that restricted feeding alters the diversity of the intestinal microbiota, increasing the abundance of *Firmicutes* and decreasing the abundance of *Bacteroidetes* and limiting the growth rate of ducks, leading to decreases in abdominal fat weight and percentage and increases in breast and leg muscle ratios. Correlation analysis revealed significant correlations between intestinal microbiota and carcass phenotypes [4]. Intestinal microbiota aid in the host-specific digestion and absorption of polyphenols, flavonoids, and other substances difficult to metabolize in the gut. Their metabolic by-products, short-chain fatty acids, can modulate the host immune response and help maintain immune homeostasis [5,6]. Collectively, these findings demonstrate the substantial role of microbial communities in the growth, development, and immune metabolism of poultry. Therefore, studying the intestinal microbiota of poultry can improve the understanding of their microbial composition, serving as a theoretical basis for research on beneficial feed additives such as probiotics and prebiotics.

*Firmicutes*, *Bacillus subtilis*, *Streptococcus*, *Lachnoclostridium*, and *Bifidobacterium* can regulate intestinal microbiota, increase the abundance of probiotics, promote nutrient digestion rates, enhance epithelial development and barrier function, facilitate the absorption and utilization of nutrients, and effectively improve the growth performance of lion-head geese [7,8,9]. An increase in the relative abundance of *Bacteroides*, *Fecalibacterium*, and *Paraprevotella* can improve the morphology of the small intestine and the microbial structure of the cecum, enhance the antioxidant and immune capabilities of Zi geese, and improve their growth performance [10]. Increasing the relative abundance of *Bacteroides*, *Escherichia–Shigella*, and *Butyricicoccus* while decreasing the relative abundance of the family *Ruminococcaceae* (*p* < 0.05) induces changes in the digestive system, significantly affects the metabolism of nutrients, and reduces the growth performance of Wanxi white geese [11]. Additionally, research has reported that bile acids can serve as effective feed additives for geese. They increase the total concentration of short-chain fatty acids (SCFAs) by enhancing the intestinal mucosal barrier, improving intestinal morphology, and altering the cecal microbiota structure, improving lipid metabolism and intestinal health in Holldobagy geese [12,13,14]. Adding 15% fermented maize stover as a partial replacement for feed during the fattening stage significantly increased the relative abundance of beneficial microorganisms, *Coprococcus* and *Victivallis*, in Xianghai flying geese. The method improved the morphological characteristics of the intestines of Xianghai flying geese and effectively maintained their production performance [15]. Moreover, when comparing the ratio of ryegrass to commercial feed added at 3:1 with full commercial feed, the ratio of ryegrass to commercial feed added at 1.5:1, and the ratio of ryegrass to commercial feed added at 2:1, cellulolytic microbes (*Ruminiclostridium* and *Ruminococcaceae UCG-010*) and lipid metabolism pathways were significantly enriched. This may have contributed to a reduction in abdominal fat accumulation in Yangzhou geese [16]. The larger the Shannon index value and the lower the Simpson index value, the higher the microbial diversity; a more diverse microbial community enhances the stability of the intestinal microbiota and resistance to pathogens [17,18]. However, feeding diets containing fiber levels of 2.5%, 6.1%, and 4.3–6.1% showed that when the fiber feeding level was 2.5%, the total OTUs, Shannon index, Chao1 index, and Simpson index of the cecal microbiota in geese were reduced. This result indicates that a low-fiber diet (2.5%) reduced the number of microbial species, microbial diversity, and microbial abundance in the cecum [19]. Although many studies have examined the critical impact of the cecal microbiota structure on goose growth and development, they were mostly conducted based on the addition of different additives or feeds. Limited research has been conducted on the intestinal microbiota of geese with significant differences in body size under the same age and feeding conditions.

Therefore, this study selected healthy Yangzhou geese of the same age and feeding method and conducted a comprehensive analysis of cecal microbiota using 16S rRNA sequencing in geese with large (L) and small (S) body weights to explore the correlation between intestinal microbiota and body weight variation in Yangzhou geese. This study contributes to a comprehensive and scientific understanding of the structure and function of goose intestinal microbiota, providing a theoretical basis for the future development of poultry probiotics and contributing to the scientific management of poultry farming and the optimization of feed formulations, effectively enhancing the productivity and quality of the poultry industry.

## 2. Materials and Methods

### 2.1. Animals and Sample Collection

The male Yangzhou geese in this experiment were obtained from Yangzhou Tiange Goose Industry Development Co., Ltd. (Yangzhou, China). A total of 300 one-day-old male Yangzhou geese were selected and raised on a standard feeding regimen. During the experiment, we measured body weight (g) at 0, 2, 4, 6, 8, and 10 weeks of age. After 10 weeks (70 d), the entire geese group was weighed. Based on the weight data, six L geese and six S geese were selected for the comparative analysis of 16S RNA intestinal microorganisms. Arterial blood samples were collected from the necks of the animals to ensure euthanasia. Slaughter performance was measured and calculated based on standard poultry production performance metrics (NY/T 823-2020 [20]) and measurement statistics. Cecal content samples were collected in a relatively sterile environment and stored in liquid nitrogen. Six intestinal samples from each group were randomly selected for 16S rRNA sequencing.

### 2.2. DNA Extraction and 16S rRNA Sequencing

The DNA was extracted from the samples using a Magnetic Soil and Stool DNA Kit (TIANGEN Biotech (Beijing) Co., Ltd., Beijing, China) following the manufacturer’s instructions. The quality and integrity of the extracted DNA were evaluated using a NanoDrop 2000 spectrophotometer and 1% agarose gel electrophoresis. For PCR amplification of the 16S rRNA gene regions 16SV3 and V4, primers 341F (5′-CCTACGGGRBGCASCAG-3′) and 806R (5′-GGACTACNNNGGGTATCTAAT-3′) were used. The PCR reactions were set up by combining 15 µL Phusion High-Fidelity PCR Master Mix (New England Biolabs, Shanghai, China), 0.2 µM primers, and 10 ng template DNA. The PCR program comprised an initial denaturation at 98 °C for 1 min, followed by 30 cycles of denaturation at 98 °C for 10 s, annealing at 50 °C for 30 s, and extension at 72 °C for 30 s. A final extension step at 72 °C for 5 min concluded the reaction. PCR products were purified using a Qiagen Gel Extraction Kit (Qiagen, Nasdaq, Germany). Sequencing libraries were generated using a TruSeq DNA PCR-Free Sample Preparation Kit (Shanghai, China). The constructed libraries were quantified with Qubit 2.0 (Beijing Nuohua Zhiyuan Technology Co., Ltd., Beijing, China), assessed for quality using an Agilent Bioanalyzer 2100 system (Beijing Nuohua Zhiyuan Technology Co., Ltd., Beijing, China), and sequenced on the Illumina NovaSeq platform (Beijing Nuohua Zhiyuan Technology Co., Ltd., Beijing, China).

### 2.3. Analysis of 16S rRNA Sequencing Data

Quantitative Insights Into Microbial Ecology 2 (QIIME2) was used to analyze the 16S microbial data obtained from sequencing by Beijing Nuohua Zhiyuan Technology Co., Ltd. (Beijing, China). The DADA2 package (Version 1.26) [21] was used for the quality control of the 16S microbial sequencing data by filtering out duplicate and low-quality sequences to obtain high-quality, clean reads. Subsequently, clustering was performed on the obtained clean reads to generate amplicon sequence variables and α-rarefaction curves. High-quality representative feature sequences were then used as references for classification annotation using the Silva 138.1 database. To comprehensively reflect the richness and diversity of microbial communities within goose cecal microbiota samples, we employed species diversity curves and species accumulation boxplots to evaluate the differences in species richness and diversity among microbial communities in each sample. Based on the obtained feature sequence results, we analyzed the shared and unique feature sequences among different groups and depicted them in a Venn diagram. The α-diversity of cecal samples was estimated using four metrics: total OTU, Chao1, Shannon, Pieulou, and Simpson indices. β-diversity, assessed using unweighted UniFrac [22,23] distances in QIIME2 [24,25], was employed to evaluate differences in community structure among different samples or groups through principal coordinate analysis (PCoA) [26]. We selected the top 10 species with the highest relative abundances at the phylum and genus levels for each sample or group and plotted relative abundance bar charts at different taxonomic levels to further explore community biomarkers between groups. Based on species annotation and abundance information at the kingdom and genus levels across all samples, we selected the top 22 phyla and 35 genera, clustered them at the species level, and depicted them in a heatmap to understand the degree of aggregation of microbial communities in each sample. Finally, we used the link package to conduct a correlation analysis between the cecal microbiota and significant performance indicators (breast muscle weight, leg muscle weight, hatching weight, and 70-day-old weight) between the L and S groups, identifying key microbial communities associated with growth and development.

### 2.4. Statistical Analysis

Using the agricolae package, we performed a one-way analysis of variance (ANOVA) on the body weight dataset at 0, 2, 4, 6, 8, and 10 weeks and on the species annotation and abundance information at the phylum and genus levels, focusing on the top 22 phyla and top 35 genera (*p* < 0.05, significant; *p* < 0.01, significant; and *p* < 0.001, highly significant). Visualizations were performed using GraphPad Prism software (Version 8.0).

## 3. Results

### 3.1. Analysis of Growth Performance in L and S Groups of Yangzhou Geese

To explore the relationship between the intestinal microbiota and weight traits in Yangzhou goose, we conducted a statistical analysis of the birth weight of samples from the L and S groups and the weights at 2, 4, 6, 8, and 10 weeks (Figure 1). Significant weight gain at each consecutive time point from birth to 10 weeks of age was observed in the Yangzhou goose used in the experiment. Of particular significance, a highly significant difference in weight between the L and S group samples from 8 weeks to 10 weeks of age was observed (*p* < 0.001).

To investigate the effect of goose cecal microorganisms on host body weight changes under consistent feeding conditions, we analyzed the slaughter data of 10-week-old Yangzhou geese (Table 1). Our findings revealed significant disparities in body, chest muscle, and leg muscle weights between the experimental Yangzhou geese in the L and S groups (*p* < 0.05), validating the rationale behind our classification of Yangzhou geese into these groups. These groups’ differences in body weight and muscle mass substantiated our classification criteria. No significant differences were observed in abdominal fat weight or the weight of major internal organs (gizzard, liver, heart, and spleen) between the L and S groups (*p* > 0.05). These results suggest that the observed difference in body weight between Yangzhou geese in groups L and S stems not from fat deposition but from an increase in the weight of the breast and leg muscle tissue. This outcome lays the groundwork for subsequent analyses of microbial composition and function among different body weights in groups L and S.

### 3.2. Overview of Cecal Microbial Data in Groups L and S

To comprehensively understand the differences in the intestinal microbiota of Yangzhou geese during their growth, we used high-throughput 16S rRNA sequencing to conduct a microbial diversity study on the cecal contents of groups L and S. In total, 1,288,660 raw reads were obtained, with an average of 107,388 raw reads per sample. After quality control, Q20 exceeded 98%, and Q30 exceeded 94% for all sequence data (Appendix A). A feature metadata table was constructed for all samples, resulting in 2295 feature sequences; among these, there were 787 feature sequences in Group L and 600 in Group S, with 908 feature sequences shared between the two groups (Figure 2A). The Shannon and Chao1 indices were used to assess the average rarefaction curves of all samples (Figure 2B,C). The rarefaction curves approached stability at the end, suggesting sufficient sequencing depth in all samples to effectively reflect the richness and diversity of the microbiota in each sample. Statistical analysis of the microbial communities in the cecal contents of groups L and S revealed significant differences between the two groups (Figure 2D,E). Additionally, with an increase in sample size, the positions of the boxplots increased sharply before gradually leveling off, indicating ample species richness and adequate sampling in the sample communities. In summary, these findings indicated reliable sequencing results, facilitating subsequent analyses (Figure 2F).

### 3.3. Analysis of Cecal Microbial Diversity and Composition in Groups L and S

To assess the α diversity of intestinal microbiota in groups L and S of geese, we measured the differences in α diversity between the two groups by comparing four metrics: total OTUs, Chao1, Shannon, Pieulou, and Simpson indices (Appendix A). The results indicated significant differences in the total OTUs and Chao1 indices between the intestinal microbiota of the two groups (*p* < 0.05), and no significant differences were observed in the Shannon and Simpson indices, suggesting substantial differences in abundance distribution and quantity between the two groups, with relatively minor differences in microbial diversity that did not reach statistical significance (Figure 3A). Additionally, principal coordinate analysis (PCoA) based on weighted Unifrac (left) and unweighted Unifrac (right) distances was employed to evaluate β diversity. The L and S groups exhibited distinct separations, indicating significant differences between the intestinal microbiota communities of the two groups (Figure 3B). To further explore the specific species composition of the goose intestinal microbiota, we analyzed the bacterial community composition at the phylum and genus levels in groups L and S. At the phylum level (Figure 3C), *Firmicutes*, *Bacteroidetes*, and *Proteobacteria* were predominant. *Firmicutes* (44.6%) and *Proteobacteria* (4.5%) exhibited higher relative abundance in group L than in group S, and *Bacteroidetes* (52.8%) showed higher relative abundance in group S than in group L (Appendix A). At the genus level (Figure 3D), *Bacteroides*, *Prevotellaceae_Ga6A1_group*, and *Subdoligranulum* were the predominant genera. We identified that the two groups had a highly abundant genus, *Prevotella*, in common. *Prevotella* had an abundance of 22.4% in group L and 31% in group S. *Prevotellaceae_Ga6A1_group* (8.4%) and *subdoligranulum* (5.3%) exhibited higher abundance in group L (Appendix A). To explore the differences in the composition and structure of the intestinal microbiota between groups L and S, we conducted a one-way ANOVA for microbial communities at the *genus* and *phylum* levels. Using the Agricolae package, we identified six significantly different microbes between groups L and S at the *genus* level, and no significant differences were observed at the *phylum* level (Figure 3E,F). *Colidextribacter* (*p* = 0.0278), *unidentified_Oscillospiraceae* (*p* = 0.0203), *Negativibacillus* (*p* = 0.0232), and *[Ruminococcus]_torques_group* (*p* = 0.0261) exhibited significant differences in abundance between the two groups, and *Prevotellaceae_Ga6A1_group* (*p* = 0.00828) and *Rikenellaceae_RC9_gut_group* (*p* = 0.00138) showed significant differences between groups L and S.

### 3.4. Correlation Analysis of Cecal Microbiota and Weight-Related Traits in Groups L and S

To investigate the correlation between cecal microbiota and body weight in Yangzhou geese, we utilized the linkET package to conduct Spearman correlation analysis of the cecal microbiota of the L and S groups and productivity indicators with significant differences between the two groups (breast muscle weight, leg muscle weight, birth weight, and weight at 70 d of age). At the phylum level, *Bacteroidetes* and *Firmicutes* showed a significant negative correlation (*R* = −0.96, *p* < 3.7831 × 10^−7^), and *Actinobacteriota* and *Bacteroidota* exhibited a significantly negative correlation (*R* = −0.58, *p* < 0.046). Mantel test correlation analysis indicated a significant positive correlation between birth weight and *Deferribacterota* (*R* = 0.77, *p* = 0.003) (Figure 4A). At the genus level, *[Ruminococcus]_torques_group* and *Prevotellaceae_Ga6A1_group* showed a significant negative correlation (*R* = −0.68, *p* < 0.013), and *Subdoligranulum* and *Erysipelatoclostridium* exhibited a significant positive correlation (*R* = 0.81, *p* < 0.0012). Mantel test correlation analysis revealed a significant positive correlation between breast muscle weight and *Prevotellaceae_Ga6A1_group* (*R* = 0.57, *p* = 0.002), leg muscle weight and *Prevotellaceae_Ga6A1_group* (*R* = 0.75, *p* = 0.001), and *Prevotellaceae_Ga6A1_group*, *[Ruminococcus]_torques_group*, and weight at 70 d of age (Figure 4B). 

## 4. Discussion

The intestinal microorganisms of poultry form a large and dynamically changing community that creates a mutualistic symbiotic relationship with the host. Many studies have demonstrated that these microorganisms play crucial roles in host digestion, energy metabolism, immune function regulation, growth, and development [7,10,12]. Although many studies have examined the key factors influencing the cecal microbial community structure in the growth and development of Yangzhou geese, most have focused on the addition of different additives or feeds. However, little is known about the intestinal flora of geese with distinct body shape differences. This study compared birth weight and body weight data at 2, 4, 6, 8, and 10 weeks for the L and S Yangzhou geese, using the same management model for statistical analysis. We observed that the weight of Yangzhou geese in group L was consistently higher than that in group S. Additionally, slaughter performance measurements revealed significant differences in chest muscle weight and leg muscle weight between group L and group S (*p* < 0.05), and no significant differences were observed in abdominal fat weight or the weights of major internal organs (gizzard, liver, heart, and spleen) (*p* > 0.05). These results indicate that the significant difference in body weight between groups L and S is not due to fat deposition but an increase in the weight of the chest and leg musculature.

We performed α-diversity and β-diversity analyses on the intestinal microorganisms of groups L and S. Alpha diversity analysis typically uses four indicators: total OTU, Shannon index, Simpson index, and Chao1 index. The total OTU represents the richness of microbial species, the Shannon and Simpson indices represent the diversity of microorganisms in the sample, and the Chao1 index reflects the abundance distribution diversity of microbial groups [19,27,28]. The results showed significant differences in the abundance and quantity of microbiota between the two groups, and the differences in microbial diversity did not reach significance. Beta diversity was evaluated using principal coordinate analysis. Groups L and S showed a clear separation, indicating substantial differences between their intestinal microbiota [29,30]. We further analyzed the microbial composition at the *phylum* and *genus* levels. At the *phylum* level, this study found that *Firmicutes*, *Bacteroidetes*, and *Desulfobacteria* were the main bacterial groups. This research revealed that supplementation with honeycomb flavonoids (HF) significantly increased the abundance of Firmicutes and some probiotics (*Clostridiales*, *Streptococcus*, *Lachnoclostridium,* and *Bifidobacterium*), promoting nutrient absorption and utilization and effectively improving the growth performance of Lionhead geese. The increased relative abundance of *Bacteroidetes*, *Faecalibacteria*, and *Paraprevotella* effectively improved jejunal morphology and cecal flora structure, thereby enhancing the growth performance of Ziyan geese [10]. However, increasing the relative abundance of *Bacteroidetes*, *Shigella*, and *Dinomycetes* while decreasing the relative abundance of *Ruminococcaceae* (*p* < 0.05) seriously affected nutrient metabolism in Wanxi White Geese, reducing their growth performance [11].In summary, the increase in *Firmicutes* and *Bacteroidetes* has a significant impact on the growth performance of geese. In this study, *Firmicutes* (44.6%) showed a higher relative abundance in the L group, while *Bacteroidetes* (52.8%) showed a higher relative abundance in the S group. The *Firmicutes*/*Bacteroidetes* ratio has been shown to be significantly correlated with gut microbiota status signaling, and an increase in this ratio is directly associated with improved growth performance [31,32]. The *Firmicutes*/*Bacteroidetes* ratio in the L group was about 20% higher than that in the S group, indicating that this ratio plays a direct role in the growth and development of geese, which directly affects the growth and development of the geese in the S group. Additionally, Spearman’s correlation analysis revealed a highly significant negative correlation between the abundance of *Bacteroidetes* and *Firmicutes*. The functions of the intestinal microorganisms can be further elucidated at the *genus* level. *Bacteroides* play a crucial role in helping the intestines decompose sugars, improving utilization and immunity [33,34], and *Prevotellaceae* help decompose sugars into SCFAs, protect the intestines, and inhibit inflammation [35,36,37]. In this study, *Bacteroides* and *Prevotellaceae_Ga6A1_group* were the dominant genera. The microbiota with the highest content in both groups L and S was *Bacteroidetes*, and *Prevotellaceae_Ga6A1_group* showed a higher abundance in group L than in group S. We inferred that the relative abundance of *Prevotellaceae_Ga6A1_group* in group S was lower than that in group L, leading to a decreased conversion efficiency of sugars to SCFAs, which indirectly affects the growth performance of geese.

In this study, a single-factor ANOVA was performed on the composition and structure of the microbiota at the *genus* and *phylum* levels. Six different microorganisms were identified in groups L and S, all of which showed significant differences at the *genus* level. The abundance of *Olidextribacter*, *unidentified_Oscillospiraceae*, *Negativibacillus*, and *[Ruminococcus]_torques_group* differed significantly between the two groups, and the *Prevotellaceae_Ga6A1_group* and *Rikenellaceae_RC9_gut_group* showed extremely significant differences. Spearman correlation analysis was conducted on the cecal microbiota of groups L and S and production performance indicators, with significant differences between the two groups (breast muscle weight, leg muscle weight, birth weight, and 70-day-old weight). This analysis identified the *Deferribacterota phylum* and *Prevotellaceae_Ga6A1_group* as two microbial groups potentially related to geese growth and development. At the *phylum* level, hatching weight is significantly and positively correlated with Deferribacterota abundance. At the *genus* level, leg and breast muscle weights were significantly positively correlated with the *Prevotellaceae_Ga6A1_group*. Additionally, a significant negative correlation was observed between *[Ruminococcus] torque* and *PrevotellaceaeGa6A1*.

In summary, the *Prevotellaceae_Ga6A1_group genus* showed a significant difference in colony abundance between groups L and S, playing a critical role in promoting the growth and development of leg and breast muscles in geese. Conversely, a high abundance of *[Ruminococcus]_torques_group* may be detrimental to the growth and development of these muscle groups. Inspection of the genus abundance of the *[Ruminococcus] torques group* revealed that its abundance in group L was lower than that in group S (Appendix A).

## 5. Conclusions

This study conducted a comprehensive analysis of the cecal flora of L and S Yangzhou geese (70 days old) using 16S rRNA sequencing to explore the intrinsic link between gut microbes and body weight. *Genus Prevotellaceae_Ga6A1_group* had extremely significant differences in colony abundance between groups L and S and showed a significant positive correlation with chest and leg muscle weight. Additionally, the *[Ruminococcus]_torques group* and the *Prevotellaceae_Ga6A1_group* showed a significant negative correlation. In summary, *Prevotellaceae_Ga6A1_group* plays a critical role in promoting the growth and development of Yangzhou goose leg and chest muscles. Conversely, a high abundance of *[Ruminococcus]_torques_group* may be detrimental to the growth and development of these muscle groups.

## Figures and Tables

**Figure 1 animals-14-02042-f001:**
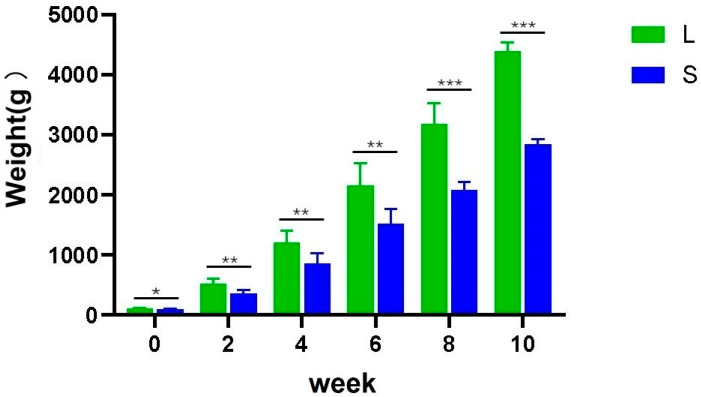
Body weight changes in groups L and S are depicted in the graph. The *x*-axis represents time changes (0 represents birth weight, 2, 4, 6, 8, and 10 represent weeks of age); the *y*-axis represents body weight changes (in grams). * indicates a difference between groups (*p* < 0.05); * *p* < 0.05 indicates differences; ** *p* < 0.01 indicates significant differences; *** *p* < 0.001 indicates extremely significant differences.

**Figure 2 animals-14-02042-f002:**
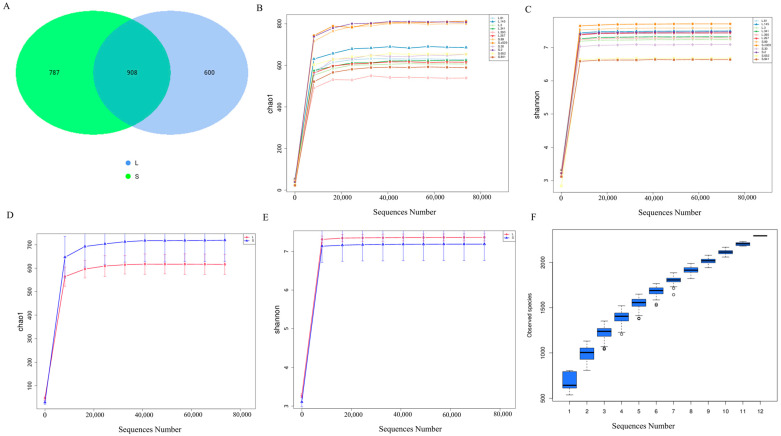
Sample richness and diversity statistics. (**A**) Characteristics of sequence statistics for groups L and S. (**B**–**E**) Dilution curves. (**B**,**C**) Richness and diversity statistics for all samples, with different colors representing different samples. (**D**,**E**) Overall richness and diversity statistics for groups L and S. (**F**) Species accumulation box plot, with sample size on the *x*-axis and the number of characteristic sequences after sampling on the *y*-axis.

**Figure 3 animals-14-02042-f003:**
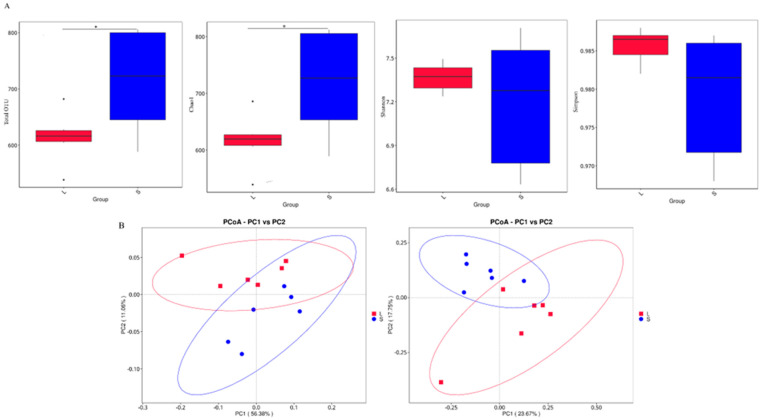
Differences in cecal microbial diversity and composition between groups L and S. (**A**) Box plot illustrating inter-group differences in alpha diversity indices. (**B**) PCoA plot. Left: weighted UniFrac; right: unweighted UniFrac. (**C**) Relative abundance of taxa at the phylum level (top 10). (**D**) Relative abundance of taxa at the genus level (top 10). (**E**) Heatmap showing the major phyla of cecal microbiota. (**F**) Heatmap showing the major genera of the cecal microbiota. * *p* < 0.05 indicates differences between groups L and S; ** *p* < 0.01 indicates significant differences between groups L and S.

**Figure 4 animals-14-02042-f004:**
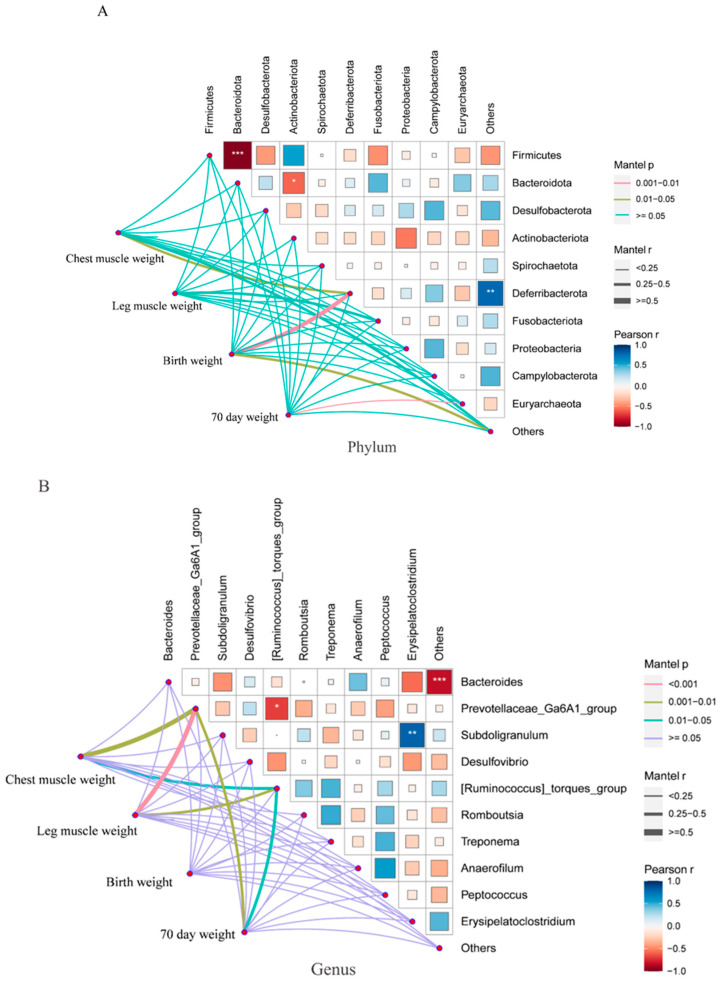
Correlation analysis of gut microbiota and slaughter performance in Yangzhou geese. (**A**) Association analysis at the phylum level between cecal microbiota and breast muscle weight, leg muscle weight, birth weight, and weight at 70 d old in Yangzhou geese. (**B**) Association analysis at the genus level between cecal microbiota and breast muscle weight, leg muscle weight, birth weight, and weight at 70 d old in Yangzhou geese. * *p* < 0.05 indicates differences; ** *p* < 0.01 indicates significant differences; *** *p* < 0.001 indicates extremely significant differences.

**Table 1 animals-14-02042-t001:** Slaughter measurements of 10-week-old Yangzhou geese.

Tissue	S Group	L Group
Heart weight (g)	26.32 ± 2.82	23.64 ± 4.07
Liver weight (g)	70.26 ± 12.75	65.24 ± 9.45
Spleen weight (g)	5.57 ± 1.79	3.93 ± 0.64
Gizzard net weight (g)	159.17 ± 17.41	148.67 ± 15.18
Abdominal fat weight (g)	60.11 ± 23.16	41.21 ± 9.30
Leg muscle weight (g)	342.7 ± 15.77 b	377.6 ± 19.61 a
Breast muscle weight (g)	233.7 ± 21.50 b	278 ± 32.26 a
Birth weight (g)	95.67 ± 4.11 ^b^	112.17 ± 11.33 ^a^
70-day-old weight (g)	2841.67 ± 72.90 ^B^	4390 ± 130.51 ^A^

Note: Different lowercase letters indicate significant differences (*p* < 0.05); different uppercase letters indicate extremely significant differences (*p* < 0.01). Absence of letters indicates no significant difference (*p* > 0.05).

## Data Availability

All data generated or analyzed during this study are included in this published paper.

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
