# Peer review of "A 16S RNA Analysis of Yangzhou Geese with Varying Body Weights: Gut Microbial Difference and Its Correlation with Body Weight Parameters"

_animals, 2024, doi:10.3390/ani14142042_

Round 1
Reviewer 1 Report
Comments and Suggestions for Authors
The authors investigated the cercal microbiota of Yangzhou geese with different body weights using 16S rRNA sequencing. They presented interesting data that showed Prevotellaceae_Ga6A1_contributed to the growth and development of goose legs and breast muscles. Thus, the study might provide insights into the role of probiotics in improving poultry growth performance. In general, the experimental design is reasonable, the data is reliable, and it demonstrates a certain degree of novelty. Even though the manuscript had some interesting data, the following are some other major observations that need the authors' clarification. In addition, the manuscript was poorly written, it is strongly recommended that a native English speaker should perform editing of the manuscript.
These include but are not limited to, some of the following issues:
1. Line 49, Change " PrevotellaceaeGa6A1" to " Prevotellaceae_Ga6A1".
2. Line 78-82, Line 90-93, More references need to be supplemented and simplify the sentence content
4. Line 126-127: Check the font.
5. Line 128-129, Rivise the sentence as follows, “During the experiment, we measured body weight (g) at 0, 2, 4, 6, 8, and 10 weeks of age”.
6. Line 137, The title "2.2 DNA extraction and 16S rRNA sequencing," should be consistent with the formatting of the other titles.
7. Line 138, Change “DNA” to “The DNA”, and line 143 “PCR” to “The PCR”.
8. Line 143, Keep the font consistent
9. Line 158, Change “Clean Reads” to “clean reads” and conduct a comprehensive genetic check for the entire article.
10. Line 190, I suggest replacing “geese” with “goose”.
11. Table 1, There should be no space before "significance ab."
12. Line 216-218, Delete "Table 1. Slaughter measurements of 10-week-old Yangzhou geese.", and check the font size of the "P" format in the entire text is consistent.
Comments on the Quality of English LanguageThe English writing in this manuscript is generally clear and comprehensible, effectively conveying the scientific findings. However, there are some areas where the language could be further polished to enhance readability and precision. Occasional grammatical errors and awkward phrasings slightly detract from the overall quality of the writing. Improving these aspects will help clearly communicate the research to a broader audience. Additionally, ensuring consistent use of technical terminology and maintaining a formal academic tone throughout the manuscript will strengthen its impact.
Author Response
Dear reviewer #1:
- Line 49, Change " PrevotellaceaeGa6A1" to " Prevotellaceae_Ga6A1".
Response: We gratefully appreciate for your valuable suggestion. We have replaced “PrevotellaceaeGa6A1” with “Prevotellaceae_Ga6A1_group”. (Line 50, page 2)
- Line 78-82, Line 90-93, More references need to be supplemented and simplify the sentence content
Response: Thank you very much for the positive comments and constructive. We have revised this section in accordance with the feedback provided by the reviewer.
We have added the corresponding references.
Firmicutes, Bacillus subtilis, Streptococcus, Lachnoclostridium, and Bifidobacterium can regulate intestinal microbiota, increase the abundance of probiotics, promote nutrient digestion rates, enhance epithelial development and barrier function, facilitate the absorption and utilization of nutrients, and effectively improve the growth performance of lion-head geese[7-9]. (Line 82, page 2)
- Odenwald, M.A.; Turner, J.R. The intestinal epithelial barrier: a therapeutic target? Nature reviews. Gastroenterology & hepatology 2017, 14, 9-21, doi:10.1038/nrgastro.2016.169.
- Hemarajata, P.; Versalovic, J. Effects of probiotics on gut microbiota: mechanisms of intestinal immunomodulation and neuromodulation. Therapeutic advances in gastroenterology 2013, 6, 39-51, doi:10.1177/1756283x12459294.
They increase the total concentration of short-chain fatty acids (SCFAs) by enhancing the intestinal mucosal barrier, improving intestinal morphology, and altering cecal microbiota structure, improving lipid metabolism and intestinal health in Holldobagy geese[12-14]. (Line 93, page 2)
- Zhang, Y.; Zhou, N.; Wu, J.; Song, L.; Bao, Q.; Weng, K.; Zhang, Y.; Vongsangnak, W.; Chen, G.; Xu, Q. Gut microbiome and serum metabolome analyses identify Bacteroides fragilis as regulators of serotonin content and PRL secretion in broody geese. Journal of Integrative Agriculture 2024, 23, 2033-2051, doi:https://doi.org/10.1016/j.jia.2023.12.031.
- Rowland, I.; Gibson, G.; Heinken, A.; Scott, K.; Swann, J.; Thiele, I.; Tuohy, K. Gut microbiota functions: metabolism of nutrients and other food components. European journal of nutrition 2018, 57, 1-24, doi:10.1007/s00394-017-1445-8.
- Line 126-127: Check the font.
Response: Thank you for pointing out this problem in this manuscript. We have checked and modified.
- Line 128-129, Rivise the sentence as follows, “During the experiment, we measured body weight (g) at 0, 2, 4, 6, 8, and 10 weeks of age”.
Response: Thank you for pointing out this problem in this manuscript. We have modified the sentence to “During the experiment, we measured body weight (g) at 0, 2, 4, 6, 8, and 10 weeks of age”. (Lines 127-128, page 3)
- Line 137, The title "2.2 DNA extraction and 16S rRNA sequencing," should be consistent with the formatting of the other titles.
Response: Thank you for pointing out this problem in this manuscript. We have checked and modified. (Line 136, page 3)
- Line 138, Change “DNA” to “The DNA”, and line 143 “PCR” to “The PCR”.
Response: We feel sorry for the inconvenience brought to the reviewer. We have made change “DNA” to “The DNA”, and “PCR” to “The PCR”. (Lines 137-142, page 3)
- Line 143, Keep the font consistent
Response: We feel sorry for the inconvenience brought to the reviewer. We have reviewed and revised the document to maintain a consistent format. (Line 142, page 3)
- Line 158, Change “Clean Reads” to “clean reads” and conduct a comprehensive genetic check for the entire article.
Response: We gratefully appreciate for your valuable suggestion. We have Change “Clean Reads” to “clean reads”. (Line 157, page 4)
- Line 190, I suggest replacing “geese” with “goose”.
Response: Thank you for pointing out this problem in manuscript. We have Change “geese” to “goose”. (Lines 189-192, page 4)
- Table 1, There should be no space before "significance ab."
Response: We feel sorry for the inconvenience brought to the reviewer. We have revised the text to maintain consistent formatting. (Line 214, page 5)
- Line 216-218, Delete "Table 1. Slaughter measurements of 10-week-old Yangzhou geese.", and check the font size of the "P" format in the entire text is consistent.
Response: We gratefully appreciate for your valuable comment. We have checked and revised “Table 1: Measurements of 10-week-old Yangzhou geese at slaughter” and ensured the font size of the “P” format is consistent throughout the text. (Lines 215-217, page 5)

Reviewer 2 Report
Comments and Suggestions for Authors
The authors designed a high-low body weight model for this goose species, compared the gut microbiota, and analyzed their correlation. The manuscript requires improvements in grammar, clarity, and critical analysis. The study addresses a relevant topic and provides some valuable data, but it could be strengthened by a more detailed analysis and a more critical discussion.
Title: Microbial Different should be Microbial Difference.
For the whole body, please italicize the Family, Genus level.
Line 20: why Correlation is with capital C?
The abstract provides a clear summary but could benefit from a more detailed description of the methods and statistical analysis used.
Line 40&41: phylum level and genus level both used Bacteroidetes, please clarify.
Line 223: Table 2 NOT Table 1
Line 275 & 276: Phylum and Genus levels are backwards
Table 3&4 are showing the same information as Figure 3C&D
Comments on the Quality of English Language
The English is easy to read and understand for most of the parts. Minor improvement is needed.
Author Response
Dear reviewer #2:
Title: Microbial Different should be Microbial Difference.
Response: We gratefully appreciate for your valuable comment. We modified “Microbial Different” to “Microbial Difference”. (Line 3, page 1)
For the whole body, please italicize the Family, Genus level.
Response: We gratefully appreciate for your valuable comment. We have italicized the family and genus names throughout the text.
Line 20: why Correlation is with capital C?
Response: We feel sorry for the inconvenience brought to the reviewer. Due to a writing error, we have now made a correction and re-edited the sentence to ensure it makes sense. (Line 20, page 1)
The abstract provides a clear summary but could benefit from a more detailed description of the methods and statistical analysis used.
Response: We gratefully appreciate for your valuable suggestion. we have added a related analysis method. (Lines 34-39, page 1)
“In this study, the cecal microbiota of healthy Yangzhou geese with large (L) and small (S) body weights, all at the same age (70 days old) and under the same feeding conditions, were sequenced using 16S rRNA. The sequencing results were annotated using QIIME2 (classify-sklearn algorithm) software, and the linkET package was used to explore the correlation between intestinal microor-ganisms and the body weight of Yangzhou goose (Spearman).”
Line 40&41: phylum level and genus level both used Bacteroidetes, please clarify.
We feel sorry for the inconvenience brought to the reviewer. Due to our writing errors and the bad reading experience we brought to you, we have replaced “Bacteroidetes” with “Bacteroides”. (Line41, page 1)
Line 223: Table 2 NOT Table 1
Response: We feel sorry for the inconvenience brought to the reviewer. We have replaced “Table S 1” with “Table 1”. (Line 224, page 6)
Line 275 & 276: Phylum and Genus levels are backwards
Response: We feel sorry for the inconvenience brought to the reviewer. We apologize for our writing error and have now corrected it.
Table 3&4 are showing the same information as Figure 3C&D
Response: We feel sorry for the inconvenience brought to the reviewer. We have moved Table 3 & 4 to the Supplementary Materials

Reviewer 3 Report
Comments and Suggestions for Authors
Dear Authors,
the topic you have proposed is interesting for the experts in the poultry sector, especially considering that for minor avian species (such as geese) there is a lack of literature compared with broilers and laying hens. The manuscript, entitled "A 16S RNA Analysis of Yangzhou Geese with Varying Body Weights: Gut Microbial Different and Its Correlation with Body Weight", provides new knowledge on the composition of gut microbiota in goose and tries to correlate the 16S metabarcoding results with different performance parameters. Overall, the manuscript has good scientific merit, the introduction and methods are good and quite exhaustive. Differently, the discussion and conclusion need some improvements. Therefore, in my opinion, some points need to be clarified and improved before considering this manuscript deserving of publication in Animals Journal. Please you can find my comments here below.
General comments:
1. Please report in Italics all bacterial species and other taxa, as required by the scientific nomenclature. Ex. : line 56: “Lactobacillus, Blautia bifidobacterium, …”
2. Improve the English overall in the manuscript, especially in the simple summary and the abstract. For example, the adverb “THUS” is used too many times in these sections and some sentences do not sound clear and correct. Please check it very carefully.
Title
1. I suggest adding the word “parameters” at the end of the title.
Simple summary
1. Line 17: Remove “Thus”.
2. Line 20-23: this sentence is not clear, please make it more clear and correct.
Material and Methods
1. Please specify also in the materials and methods that this study has been conducted in compliance with current regulations in your country and report the approval number.
2. Lines 131-132: please report the specific standard poultry metrics and statistics that you used, possibly with references.
3. Were cecal samples collected in an aseptic way? Please clarify this point and add it to the text.
4. Please explain why you selected only 6 samples from the 2 groups. The sample size is really small and usually in this kind of study sample size is bigger. In addition, the initial total number of geese is 300 and 12 samples did not seem a representative value for this work.
5. Did you conduct the cutadapt analysis on raw reads? Normally, for 16S metabarcoding analysis primers must be removed from the reads before proceeding with DADA2 analysis.
6. Please add, add also the pieulou’s evenness alpha diversity metric. It can be easily obtained in QIIME2.
Results
1. Move Table 2 to supplementary materials.
2. Table 3 and Table 4: please specify if these data are absolute or relative values.
Discussion
The discussion lacks of comparison with the published literature. Please compare your 16S metabarcoding results with already published data on the goose. At least, compare your results with these two studies reporting the effects of overfeeding on the geese gut microbiota.
https://pubmed.ncbi.nlm.nih.gov/33652539/
https://pubmed.ncbi.nlm.nih.gov/34102480/
Comments on the Quality of English LanguageImprove the English overall in the manuscript, especially in the simple summary and the abstract. For example, the adverb “THUS” is used too many times in these sections and some sentences do not sound clear and correct. Please check it very carefully.
Author Response
Dear reviewer #3:
General comments:
- Please report in Italicsall bacterial species and other taxa, as required by the scientific nomenclature. Ex: line 56: “Lactobacillus, Blautia bifidobacterium, …”
Response: We feel sorry for the inconvenience brought to the reviewer. We have already followed your suggestionsreport in Italics all bacterial species and other taxa, as required by the scientific nomenclature.
- Improve the English overall in the manuscript, especially in the simple summary and the abstract. For example, the adverb “THUS” is used too many times in these sections and some sentences do not sound clear and correct. Please check it very carefully.
Response: We gratefully appreciate for your valuable comment. We have reviewed and revised the entire article, reducing the use of adverbs to ensure sentence clarity.
Title
- I suggest adding the word “parameters” at the end of the title.
Response: We gratefully appreciate for your valuable comment. We've added the word "Parameters" to the end of the title as you suggested. (Line 4, page 1)
Simple summary
- Line 17: Remove “Thus”.
Response: Thank you for pointing out this problem in manuscript. We have removed “Thus” as suggested by the reviewer. (Line 17, page 1)
- Line 20-23: this sentence is not clear, please make it more clear and correct.
Response: We feel sorry for the inconvenience brought to the reviewer. We have made correction according to the Reviewer’s comments.
“Correlation analysis of production performance indicators between cecal microflora and body weight revealed a significant positive correlation between hatching weight and Deferribacterota at the phylum level.” (Lines 20-22, page 1)
Material and Methods
- Please specify also in the materials and methods that this study has been conducted in compliance with current regulations in your country and report the approval number.
Response: We gratefully appreciate for your valuable suggestion. we based this study on performance terminology and measurements for poultry (NY/T 823-2020), and we have included the relevant report approval number in the text. (Line 133, page 3)
- Lines 131-132: please report the specific standard poultry metrics and statistics that you used, possibly with references.
Response: We gratefully appreciate for your valuable suggestion. Our study is based on the People's Republic of China Agricultural Industry Standard Poultry Production Performance Terminology and Measurement Calculation Methods, approval number NY/T 823-2020.
- Were cecal samples collected in an aseptic way? Please clarify this point and add it to the text.
Response: We gratefully appreciate for your valuable suggestion. we created a relatively sterile environment and used liquid nitrogen for rapid freezing throughout the sampling process. The short freezing time prevented bacterial growth, providing double protection.
- Please explain why you selected only 6 samples from the 2 groups. The sample size is really small and usually in this kind of study sample size is bigger. In addition, the initial total number of geese is 300 and 12 samples did not seem a representative value for this work.
- Response: We feel sorry for the inconvenience brought to the reviewer. We chose to raise 300 Yangzhou geese under the same feeding method, management method, and environment to ensure individuals could be randomly selected from a large group, reducing variability and production errors, and thus ensuring experimental accuracy. Later, we divided the geese into two groups based on their weight and randomly selected 12 individuals from group for sequencing. This random sampling throughout the process minimized experimental errors. By analyzing the sequencing results of the 12 samples, we found the results were consistent with our expectations. Thank you for your suggestions; we will consider increasing the sample size in subsequent experiments.
- Did you conduct the cutadaptanalysis on raw reads? Normally, for 16S metabarcoding analysis primers must be removed from the reads before proceeding with DADA2
Response: We gratefully appreciate for your valuable suggestion.
YES. After truncating the barcode and primer sequences, the software FLASH (Version 1.2.11, http://ccb.jhu.edu/software/FLASH/) was used to splice the reads of each sample. The resulting spliced sequence was the raw tags data (Raw Tags). Cutadapt analysis was then performed using Cutadapt software to match the reverse primer sequence and remove the remaining sequence to prevent interference with subsequent analysis.
- Please add, add also the pieulou’s evenness alpha diversity metric. It can be easily obtained in QIIME2.
Response: We gratefully appreciate for your valuable suggestion. We have added the pieulou uniformity and alpha diversity metrics in table S2
Results
- Move Table 2 to supplementary materials.
Response: We feel sorry for the inconvenience brought to the reviewer. We have moved Table 2 to the Supplementary Materials
- Table 3 and Table 4: please specify if these data are absolute or relative values.
Response: We gratefully appreciate for your valuable comment. The average relative abundance values we used have been added to the text.
Discussion
The discussion lacks of comparison with the published literature. Please compare your 16S metabarcoding results with already published data on the goose. At least, compare your results with these two studies reporting the effects of overfeeding on the goose gut microbiota.
Response: We gratefully appreciate for your valuable comment. We have made changes based on your suggestions.
“At the phylum level, this study found that Firmicutes, Bacteroidetes, and Desulfobacteria were the main bacterial groups. The research revealed that supplementation of honeycomb flavonoids (HF) significantly increased the abundance of Firmicutes and some probiotics (Clostridiales, Streptococcus, Lachnoclostridium and Bifidobacterium), promoting nutrient absorption and utilization, and effectively improving the growth performance of Lionhead geese. The increased relative abundance of Bacteroidetes, Faecalibacteria, and Paraprevotella effectively improved jejunal morphology and cecal flora structure, thereby enhancing the growth performance of Ziyan geese [10].However, increasing the relative abundance of Bacteroidetes, Shigella, and Dinomycetes while decreasing the relative abundance of Ruminococcaceae (P<0.05) seriously affected nutrient metabolism in Wanxi White geese, reducing their growth performance [11].In summary, the increase in Firmicutes and Bacteroidetes has a significant impact on the growth performance of geese. In this study, Firmicutes (44.6%) showed a higher relative abundance in the L group, while Bacteroidetes (52.8%) showed a higher relative abundance in the S group. The Firmicutes/Bacteroidetes ratio has been shown to be significantly correlated with gut microbiota status signaling, and an increase in this ratio is directly associated with improved growth performance [31,32]. The Firmicutes/Bacteroidetes ratio in the L group was about 20% higher than that in the S group, indicating that this ratio plays a direct role in the growth and development of geese, which directly affects the growth and development of the geese in the S group.” (Lines 330-348, page10)
